# Predicting postoperative delirium assessed by the Nursing Screening Delirium Scale in the recovery room for non-cardiac surgeries without craniotomy: A retrospective study using a machine learning approach

**Niklas Giesa**[1]*, **Stefan Haufe**[1,2,3,4], **Mario Menk**[5], **Björn Weiß**[5], **Claudia D. Spies**[5], **Sophie K. Piper**[1], **Felix Balzer**[1☉], **Sebastian D. Boie**[1☉]

**1** Institute of Medical Informatics, Charité – Universitätmedizin Berlin, Berlin, Germany, **2** Berlin Center for Advanced Neuroimaging (BCAN), Charité – Universitätmedizin Berlin, Berlin, Germany, **3** Mathematical Modelling and Data Analysis Department, Physikalisch-Technische Bundesanstalt Braunschweig und Berlin, Berlin, Germany, **4** Uncertainty, Inverse Modeling and Machine Learning Group, Technische Universität Berlin, Berlin, Germany, **5** Department of Anesthesiology and Operative Intensive Care Medicine (CCM, CVK), Berlin, Germany

☉ These authors contributed equally to this work.
* niklas.giesa@charite.de

**Data Availability Statement:** Due to German data protection and patient privacy laws, we cannot

## Abstract

Postoperative delirium (POD) contributes to severe outcomes such as death or development of dementia. Thus, it is desirable to identify vulnerable patients in advance during the perioperative phase. Previous studies mainly investigated risk factors for delirium during hospitalization and further used a linear logistic regression (LR) approach with time-invariant data. Studies have not investigated patients' fluctuating conditions to support POD precautions. In this single-center study, we aimed to predict POD in a recovery room setting with a non-linear machine learning (ML) technique using pre-, intra-, and postoperative data. The target variable POD was defined with the Nursing Screening Delirium Scale (Nu-DESC) $\geq$ 1. Feature selection was conducted based on robust univariate test statistics and $L_1$ regularization. Non-linear multi-layer perceptron (MLP) as well as tree-based models were trained and evaluated—with the receiver operating characteristics curve (AUROC), the area under precision recall curve (AUPRC), and additional metrics—against LR and published models on bootstrapped testing data. The prevalence of POD was 8.2% in a sample of 73,181 surgeries performed between 2017 and 2020. Significant univariate impact factors were the preoperative ASA status (American Society of Anesthesiologists physical status classification system), the intraoperative amount of given remifentanil, and the postoperative Aldrete score. The best model used pre-, intra-, and postoperative data. The non-linear boosted trees model achieved a mean AUROC of 0.854 and a mean AUPRC of 0.418 outperforming linear LR, well as best applied and retrained baseline models. Overall, non-linear machine learning models using data from multiple perioperative time phases were superior to

provide the original data. We added comprehensive summary statistics to describe our raw data. We have provided statistics across model input values as a direct download on Dryad as suggested by PLOS Digital Health 10.5061/dryad.1vhhmgr2g. For requesting the underlying raw data, please contact the ethics committee at Charité via https://ethikkommission.charite.de/.

**Funding:** The author(s) received no specific funding for this work.

**Competing interests:** The authors have declared that no competing interests exist.

traditional ones in predicting POD in the recovery room. Class imbalance was seen as a main impediment for model application in clinical practice.

## Author summary

Currently, the pathophysiology of postoperative delirium (POD) is unknown. Hence, there is no dedicated medication for treatment. Many patients who experience POD suffer from chronic mental disorders causing pressure on related family members, clinicians, and the health system. With our study, we want to detect suspected POD before onset trying to give decision support to health professionals. Vulnerable patients could be transferred to a higher level of care mitigating the risk of severe outcomes such as long-term cognitive decline. We also provide insides into clinical parameters—recorded before, during, and after the surgery—that could be adapted for reducing POD risk. Our work is openly available, developed for clinical implementation, and could be transferred to other clinical institutions.

## Introduction

Postoperative delirium (POD) as an acute state of brain dysfunction after a surgery has been found to be related to adverse long-term effects—such as increased length of hospitalization, development of dementia, and death [1–3]. Reported incidences (3–50%) vary substantially depending on the cohort definition and are elevated in major surgical cases as well as in elderly patients [4–7]. While the etiology of POD is difficult to prove conclusively, authors investigated surgical stress as a potential cause for neuro inflammation that could subsequently result in POD, especially in the elderly population [8,9]. Recent studies stress the need for an early assessment of POD onset in the recovery room enabling clinicians to improve patients' outcomes [2,4,10,11]. Assessment scores for a recovery room setting which are validated against DSM-5 criteria comprise the Confusion Assessment Method (CAM) and the Nursing Screening Delirium Scale (Nu-DESC) [10,12,13]. In contrast to the CAM, the Nu-DESC is a purely observational score that has been validated to have a sensitivity of up to 80% for scores $\geq 1$ [10,13].

Due to the high relevance in perioperative care, previous POD studies were not limited to finding predisposing factors—such as comorbidity or age—and precipitating factors—such as surgical complications or intraoperative blood loss [2,10,14,15]. Studies went further by applying multivariable prediction models. Most of them evaluated the delirium onset during hospitalization with the CAM and used a linear logistic regression (LR) technique [16–20]. Popular models by Boogaard et al. and Wassenaar et al. show good test performance as measured by the area under the receiver operating characteristics curve (AUROC = 0.75–0.89) [21–23] but diminished performance on external data (AUROC = 0.62) [24]. A few authors trained nonlinear machine learning (ML) algorithms predicting POD [4,25–27]. Xu et al. used ICD-9 encoded POD as a target variable for a deep multi-layer perceptron (MLP) architecture. Using pre- and intraoperative variables extracted from 111,888 electronic health records (EHRs) as features, the authors achieved an AUROC of 0.72 [25]. Although Xu et al. capture the fluctuating physiology in the intraoperative phase, a meta-study by Ruppert et al. highlights that most of the published prediction models use values from a single point in time [28].

Our aim was to identify patients vulnerable to suffering from POD in the recovery room. Pre-, intra-, and postoperative variables were extracted from EHRs and combined into different prognostic non-linear ML models. We used the Nu-DESC in a recovery room setting for defining POD. An automated risk assessment after the end of the surgery could help transferring vulnerable patients to specialized noise-reduced wards improving their outcome [10,29–31]. The application of prognostic models was accompanied by analyses of effect sizes and effect directions for single predictors on POD to understand risk factors in more depth.

## Methods

### Ethics statement

This study was performed under ethics approval granted by the independent ethics committee at Charité –Universitätsmedizin Berlin (vote EA4/254/21). We performed analysis on pseudonymous data. Data processing consent was obtained by a formal in-hospital treatment contract.

### Cohort and target variable

EHRs were extracted for admissions between 01/01/2017 and 12/31/2020. Patients who underwent cardiovascular or craniotomy procedures were excluded due to the increased risk of postoperative complications [1,5,17,20]. All other adult patients ($\geq$ 18 years) who were assessed with at least one Nu-DESC in the recovery room were included. The POD positive (y = 1) group consisted of surgeries on patients who were evaluated with at least one Nu-DESC score $\geq$ 1 in the recovery room [10]. If all Nu-DESC scores in the recovery room were equal to 0, the surgery was assigned to the negative group (y = 0). Each POD target variable y was calculated per surgery directly corresponding to the hospital stay and the admitted patient. Multiple surgeries during one hospitalization were treated as individual samples.

Fig 1 summarizes the inclusion criteria yielding the cohort of 61,187 patients with 69,974 hospital stays and 73,181 performed surgeries. POD incidence was 9.3%, 8.4% and 8.2% for distinct patients, hospital stays and surgeries, respectively. Table 1 displays baseline characteristics for the selected cohort. Additional characteristics are shown in Tables A and B and Fig A in S2 Appendix.

### Perioperative time phases

The hospital stays were divided into three distinct perioperative time phases. Data from the preoperative (T1) -, intraoperative (T2) -, and postoperative (T3) phase as well as time-invariant (TI) data were considered rather than focusing on one value from a single point in time. Fig 2 highlights the start—and end events for T1-T3.

For the POD prediction task, distinct time phases (T1-T3) were considered individually or combined. A different model (M1-M123) was trained and evaluated for each combination with data from assigned time phases (see Table 2). In the following, time phases and their combinations are named as T1-T123, data from TI is always included.

### Feature extraction and preprocessing

Data were extracted from the clinical information systems (CIS) of three sides at our clinical center. Based on literature review and clinical expertise [14–28], a total of 549 clinical variables were identified with respect to T1-T3 and TI in our source systems. From these variables, 375 were available for at least 10% of the included patients Variables were grouped into 14 clinical

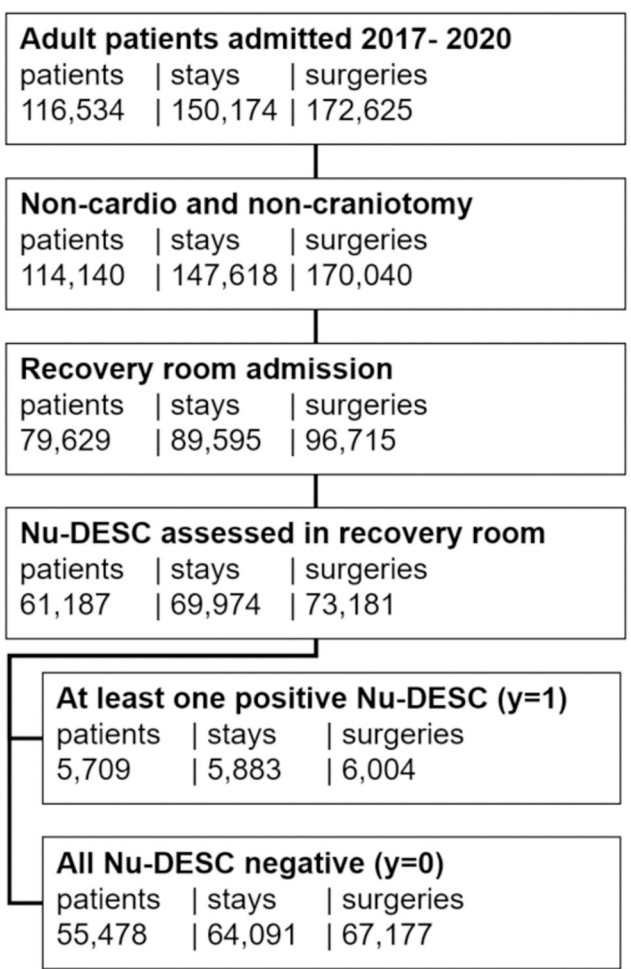

**Fig 1. Cohort definition based on inclusion criteria.** Number of patients,—hospital stays and—surgeries are provided at each step. A positive target variable y = 1 was defined based on the presence of at least one Nu-DESC score ≥ 1. A negative y = 0 was defined if all Nu-DESC scores were equal to 0.

domains such as demographics, inputs, scores and scales, or vital signs. Table 3 shows the number of extracted variables per time phase and clinical domain.

EHRs might suffer from integrity issues due to distributed CIS [32,33]. Thus, extraction, transformation, and loading (ETL) scripts were refined until there were no discrepancies with the front-end for a sample of 50 surgeries. We systematically checked these sample data for correctness and consistency with clinical experts. We integrated variables across source systems and harmonized different naming conventions as well as units. The data warehousing team at our institution provided technical documentation to understand the CIS e.g., the versioning mechanism, calculation of metrics, and data schema descriptions. Reverse engineering [34] was applied to reconstruct data that was displayed in our front-end application.

For numerical variables, valid thresholds (lower and upper bounds) were applied to remove impossible values (e.g., negative oxygen saturation) rather than omitting physiological edge cases (e.g., oxygen saturation at 65%) (see Table A in S1 Appendix). Hence, extreme values were kept initially that might contribute to the predictive performance. We preferred preserving raw data instead of heavy preprocessing. The application of robust feature aggregations

**Table 1. Baseline characteristics for all patients in the POD positive (y = 1) and negative (y = 0) groups.** The mean, [1st, 2nd, 3rd] quartiles are shown for numerical values, counts are displayed otherwise. The type of surgery is defined using clinical codes described in Table B in S1 Appendix.

| | Unit | All | POD Positives (y = 1) | POD Negatives (y = 0) |
|---|---|---|---|---|
| **Counts** | | | | |
| Patients | - | 61,187 | 5,709 | 55,478 |
| Stays | - | 69,974 | 5,883 | 64,091 |
| Surgeries | - | 73,181 | 6,004 | 67,177 |
| Surgeries per stay | - | 1.15, [1, 1, 1] | 1.15, [1, 1, 1] | 1.14, [1, 1, 1] |
| Previous admissions | - | 0.90, [0,0, 1] | 1.00, [0, 0, 1] | 0.80, [0, 0, 1] |
| Previous surgeries per stay | - | 0.51, [0, 0, 1] | 0.53, [0, 0, 1] | 0.49, [0, 0, 1] |
| **Demographics** | | | | |
| Age | years | 56, [42, 60, 73] | 60, [46, 62 76] | 52, [36, 53, 67] |
| Gender | - | 28,013 male (46%) 33,174 female (54%) | 2,927 male (51%) 2,782 female (49%) | 25,086 male (47%) 30,392 female (53%) |
| BMI | kg/m$^2$ | 26.91, [22.80, 25.70, 29.47] | 26.91, [22.85, 25.76, 29.38] | 26.91, [22.75, 25.64, 29.57] |
| ASA status | - | 2.07, [2, 2, 3] | 2.37, [2, 2, 3] | 2.04, [2, 2, 3] |
| OP N urgency class | - | 4.00, [3, 5, 5] | 3.89, [3, 5, 5] | 4.02, [3, 5, 5] |
| **Hospitalization** | | | | |
| Length of hospital stay | days | 8.66, [3.25, 5.09, 9.05] | 10.53, [3.08, 6.00, 11.00] | 6.80, [2.22, 3.78, 7.09] |
| Length of anesthesia | hours | 2.22, [1.14, 1.69, 2.54] | 2.44, [1.45, 2.14, 3.06] | 1.99, [1.12, 1.65, 2.49] |
| Length of surgery | hours | 1.28, [0.48, 0.86, 1.50] | 1.43, [0.63, 1.13, 1.88] | 1.14, [0.47, 0.84, 1.46] |
| Length of recovery room stay | hours | 2.51, [1.08, 1.59, 2.32] | 2.92, [1.45, 2.08, 3.01] | 2.11, [1.05, 1.54, 2.24] |
| **Nu-DESC evaluation** | | | | |
| Number of Nu-DESC evaluations | - | 1.09, [1, 1, 1] | 1.12, [1, 1, 1] | 1.09, [1, 1, 1] |
| Duration between recovery room admission and 1st Nu-DESC evaluation | minutes | 50.52, [7.79, 32.03, 69.22] | 35.55, [4.03, 10.84, 38.99] | 51.83, [8.67, 34.26, 71.05] |
| Duration between last Nu-DESC evaluation and recovery room discharge | minutes | 75.02, [17.97, 40.12, 80,54] | 133.33, [50.46, 93.34, 144.25] | 69.90, [17.00, 37.19, 74.15] |
| **Type of surgery** | | | | |
| Locomotive organs | - | 20,369 (37%) | 2,293 (38%) | 18,103 (27%) |
| Organs of the head | - | 12,680 (19%) | 743 (12%) | 11,937 (18%) |
| Nervous system | - | 8,205 (12%) | 1,034 (17%) | 7,171 (11%) |
| Digestive tract | - | 8,674 (13%) | 764 (12%) | 7,910 (13%) |
| Skin and tissue | - | 7,580 (11%) | 890 (15%) | 6,690 (11%) |
| Urinary system | - | 6,381 (10%) | 597 (9%) | 5,784 (9%) |
| Blood vessels | - | 3,061 (5%) | 344 (6%) | 2,717 (4%) |
| Respiratory tract | - | 3,705 (6%) | 395 (7%) | 3,310 (5%) |
| Hormone system | - | 807 (1%) | 114 (2%) | 693 (1%) |
| **Type of anesthesia** | | | | |
| General balanced | - | 36,730 (50%) | 2,819 (47%) | 33,911 (50%) |
| Total intravenous | - | 31,720 (43%) | 3,255 (54%) | 28,465 (42%) |
| Epidural | - | 2,639 (3%) | 131 (2%) | 2,508 (4%) |
| Spinal | - | 3,776 (5%) | 49 (1%) | 3,727 (6%) |
| Analgo | - | 1,145 (1%) | 91 (2%) | 1,054 (2%) |
| Other | - | 3,405 (4%) | 258 (4%) | 3,147 (5%) |

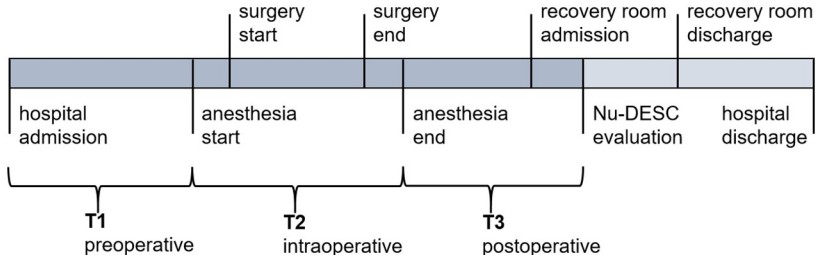

**Fig 2. Hospitalization schema with relevant intraoperative points in time.** Definition of time phases T1-T3 are based on highlighted events. TI holds time-invariant data and is not included in the graphic. When multiple Nu-DESC evaluations were performed in the recovery room, the timestamp of the first one was chosen for phases including T3.

**Table 2. Models which are fed with data from corresponding time phase combinations.** The start (from) and end (to) of each combination is introduced as well. TI data is included for all models.

| Model | Time Phases | From | To |
|---|---|---|---|
| M1 | T1 | hospital admission | anesthesia start |
| M2 | T2 | anesthesia start | anesthesia end |
| M3 | T3 | anesthesia end | 1ˢᵗ Nu-DESC evaluation in the recovery room |
| M12 | T12 (T1+T2) | hospital admission | anesthesia end |
| M23 | T23 (T2+T3) | anesthesia start | 1ˢᵗ Nu-DESC evaluation in the recovery room |
| M123 | T123 (T1+T2+T3) | hospital admission | 1ˢᵗ Nu-DESC evaluation in the recovery room |

increased the applicability of our methods. Multiple values for one surgery recorded during one time phase were aggregated as described in the next section.

## Feature encoding

Demographic characteristics—e.g., age, weight—were included without any extra feature encoding and assigned to the entire hospital stay (TI). The history of an International

**Table 3. Number of extracted variables per time phase (T1-T3, TI), and per clinical domain.**

| Clinical domain | TI | T1 | T2 | T3 | *Total* |
|---|---|---|---|---|---|
| Demographics | 5 | | - | - | 5 |
| Hospitalization | 4 | 2 | 4 | 4 | 14 |
| Comorbidities and diagnosis | 41 | 43 | 7 | 8 | 99 |
| Operation procedures | 31 | - | - | - | 31 |
| Vital signs | - | 9 | 13 | 12 | 34 |
| Respiratory | - | 4 | 9 | 5 | 18 |
| Scores and scales | - | 4 | 2 | 5 | 11 |
| EEG imaging | - | - | 7 | 7 | 14 |
| Laboratory | - | 29 | 15 | 21 | 65 |
| Input classes | - | 7 | 5 | 1 | 13 |
| Inputs | - | 2 | 22 | 18 | 42 |
| Outputs | - | - | 3 | 1 | 4 |
| Support procedures | - | 16 | 4 | 5 | 25 |
| Anesthesia procedures | - | 19 | 4 | 2 | 5 |
| *Total* | 81 | 116 | 91 | 87 | **375** |

Classification of Diseases (ICD)[35] code was summarized as the number of respective diagnoses from previous hospital stays of the same patient and assigned to TI. Hospitalization information and life support procedures—e.g., hospitalization durations, surgery durations, and ventilation durations—were calculated for each time phase (T1-T3) individually.

Numerical measurements like vital signs—e.g., respiratory rate or heart frequency–, respiratory parameters—e.g., fraction of inspired oxygen–, EEG imaging parameters—e.g., spectral edge frequencies (SEF), or patient state index (PSI)–as well as laboratory values—e.g., platelet counts in blood—were aggregated for each phase (T1-T3) by calculating the $10^{th}$, $50^{th}$ and $90^{th}$ percentiles. The volatility of each vital sign was summarized using the robust median absolute deviation (MAD) [36]. For medications, the administered amount, volume, and rate was retrieved. In addition to the calculated percentiles, the total volumes or amounts of each input —e.g., total volume of electrolyte solution–, and total volumes of each output—e.g., total volume of urine outputs—were calculated.

Information from German operation and procedure (OPS) codes [35] were used to one-hot encode different types of operation procedures—e.g., procedures on the nervous system. These classes were considered to be fixed before each hospital stay (TI). Drug classes—e.g., benzodiazepines, opioids—and types of anesthesia procedures—e.g., preoxygenation—were one-hot encoded as time-variant variables. We encoded ICD diagnoses according to their system entry time during a hospital stay as a binary variable, separately for each time phase. Concrete ICD and well as OPS codes are provided in Table B in S1 Appendix.

All categorical variables were either present (value = 1) or not present (value = 0). Thus, we could not differentiate between truly missing (undocumented) and not present (e.g., encoding that a treatment was deliberately not given) values. In contrast, such a disambiguation was possible for numerical features. A binary indicator variable was added to encode real missingness of numerical variables [37]. Additionally, we used a single imputation technique described in the standardization section. Additional details on feature encodings are provided in Table C in S2 Appendix. The selection of features per model is outlined in the next paragraphs.

## Univariate test statistics

Robust Mann-Whitney U (MWU) tests [38] were used to evaluate the discriminatory power of numerical parameters with respect to the POD target. We applied false discovery rate (FDR) correction [39] (alpha = 0.05) to identify statistically significant variables. FDR correction mitigates Type I (false-positive) errors that occur when falsely rejecting the null-hypothesis for multiple testing problems [40]. The AUROC—which can be derived from the MWU statistic [41]—was used to quantify effect-sizes. AUROC values are normalized between 0 and 1, where 1 indicates a perfect positive association, 0 indicates a perfect negative association, and 0.5 indicates chance-level discrimination. The absolute strength of an effect—regardless of the direction—was calculated for each significant variable as $e = 2|AUROC-0.5|$. For categorical variables, we used the odds ratio (OR) of a univariate logistic regression [42] as a measure of effect size. A direction independent measure of effect size was defined as $o = |log(OR)|$.

We applied two different methods to select model input features out of 375 variables that were available for at least 10% of the patients. The first method devised a feature selection algorithm to provide a tradeoff between data availability and predictive power towards the target. For numerical features, missingness ($m$) was furthermore defined as the fraction of patients having no values for more than 50% of all available features. As a measurement for the univariate predictive performance, the previously defined parameters $e$ and $o$ were used for numerical and categorical variables respectively. A grid search [43] was applied over different thresholds for $m$ ($tm = [0.4, 0.6, 0.8]$), $e$ ($te = [0.1, 0.05]$) and $o$ ($to = [1.5, 0.5]$).

Numerical features were ordered descending by their univariate performance parameter $e$. A numerical feature was drawn and added to a feature set if its value for $e$ exceeded the threshold $te$ and if the missingness in the set did not exceed the threshold $tm$. Categorical features were simply added if their values for $o$ were above the corresponding threshold $to$. When multiple time phases (T1-T3) were included for a model variant, the mean of $e$, $o$, and $m$ was calculated across these phases. In case of multiple summary statistics (e.g., percentiles) for the same feature, only the most discriminative one was ultimately included in the feature set.

The second feature selection method was performed with all available 375 features ($tm = 1$, $te = 0$, $to = 0$). We applied $L_1$-norm regularization [44] to automatically detect input variables that contribute to the POD prediction task. In contrast to the first univariate selection method, we took inter-correlation between covariates into account here. Models predicted our target and selected features in parallel. In total, we run experiments with $3 * 2 * 2 + 1 = 13$ feature sets per model variant (M1-M123) including 12 sets from the first selection method and one set from the second method. The loss calculated on a validation set was used as a cost function.

In S1 Appendix, descriptive statistics can be found in Tables C and D. A list of selected input features per model is presented in Table E.

## Data splitting, cross-validation and standardization

The extracted data were initially randomly split (80/20%) into train—and test sets. To avoid dependencies between these sets we used patient identifiers to perform the splitting. Stratification with the target variable was done so that the incidence of POD was preserved in both sets. As a result, the testing set comprised 12,238 patients, the training set included 48,949 patients. In Table C in S1 Appendix, we present descriptive statistics for both data sets indicating that the test set pose a representative sample of our training set.

The training data were used to evaluate models with different feature sets and hyperparameters. A 3-fold cross validation (CV) technique [45] was applied where a hyperparameter configuration was determined from 66.6% (2 training folds) and evaluated on 33.3% (1 holdout fold) of the training data. This evaluation was iteratively performed three times. For each model variant (M1-M123), each feature set was used in a hyperparameter search. The best performing configuration across all CV iterations per feature set was chosen on basis of the lowest validation loss for the final evaluation on the test set.

Numerical features were standardized using z-transformation [46]. Feature mean values as well as standard deviations were calculated on the training data, applied to validation (holdout folds)–and eventually to the test data. Extracted training set mean values were also used to impute missing values in train, validation, and test sets. Mean imputation was applied as calculating the mean of all values per feature across the population and replacing missing values by these feature means [47]. Additionally we encoded missingness with binary indicators [37]. Fig B in S2 Appendix shows correlations between missing indicator variables assuming that our data did not follow a missing completely at random (MCAR) pattern.

## Machine learning techniques and hyperparameter search

Two types of non-linear models were trained in comparison to linear-, and baseline models. First, deep multi-layer perceptrons (MLPs) [48,49] were trained to predict POD. Previous studies have shown that MLPs can describe highly complex non-linear functions acting as universal approximators [49]. Table D in S2 Appendix outlines the ranges of values optimized with Grid—[43] or Random Search [50] for a fully connected MLP architecture. We used focal loss [51] or weighted binary cross-entropy (BCE) [52] since they have been shown to be able to deal with unbalanced classification problems such as ours. $L_1$-norm regularization [44] was

applied on the first layer of the MLP when using all available features instead of feature subsets. Table E in S2 Appendix displays results from the CV process.

In addition to MLPs, we included two non-linear ensemble machine learning approaches based on decision trees. We selected additional ensemble techniques due to their capability of separately learning aspects of the training data and combining results into one final prediction [53]. Random forest and gradient boosting classifier were integrated into Random Search [54,55]. Table F in S2 Appendix outlines the parameter search space for tree-based models. Weighted BCE was configured for both algorithms. Table G in S2 Appendix displays results from the CV process with tree-based models.

We further compared the highly non-linear architectures with a linear logistic regression (LR) using a weighted BCE. LR models incorporated all available features per corresponding time phase (T1-T123). Constructed models (M1-M123) were also compared to LR models by Wassenaar and Boogaard [21,23]. The authors predicted delirium onset during an intensive care unit (ICU) stay assessed with the CAM. Due to the simplicity and open accessibility, we applied pre-trained models on data from time phase combinations (T1-T123). Models by Wassenaar and Boogaard were retrained and evaluated with a LR, a MLP and boosted tree technique.

The performance of the obtained predictions was assessed by means of either the AUROC or the area under the precision recall curve (AUPRC) [56]. The AUROC is less suitable—biased towards large values—for highly imbalanced classification problems such as ours. This problem is less pronounced for the AUPRC [56,57], which focuses the minority class. Additionally, the F1-score was computed [58]. To estimate standard errors of the mean model performances, bootstrapping—random sampling with replacement—was applied 1000 times on the test data [59]. By applying bootstrapping, we sought to estimate the stability of model predictions. Additionally, we read the absolute values of MLP weights from the very first $L_1$ regularized layer to give insights into input features that were focused during learning.

### Code—and data availability, reporting

The code including trained models, preprocessing scripts and usage notes, are openly accessible on our public code repository [60]. We published descriptive data for our training- and testing sets on a public Dryad repository [61]. We foster the reproducibility of our results by describing our data preprocessing pipeline and by providing guidance for applying our models. German data protection rules—that are based on the European General Data Protection Regulation (GDPR)—prohibit publicly sharing original patient data such as ours [62]. However, we present descriptive statistics of extracted and aggregated EHR in Tables C and D in S1 Appendix. Results are reported in accordance with the transparent reporting of a multivariable prediction model for individual prognosis or diagnosis (TRIPOD) guidelines (see Table I in S2 Appendix) [63].

## Results

### Perioperative variables

Univariate correlations between individual numerical as well as categorical variables and the POD target are presented in Tables 4 and 5. Highly correlated clinical variables were age (e = 0.232, with e = 2|AUROC-0.5|) for TI, the ASA (American Society of Anesthesiologists physical status classification system) status (e = 0.179) for T1, the intraoperative (T2) amount of remifentanil (e = 0.200), and the Aldrete score (e = 0.347) measured in the recovery room (T3). The anesthesia-, and the surgery durations calculated for each timeline are highly discriminative in both, the intraoperative—(T2) (anesthesia duration e = 0.218, surgery duration

**Table 4. Ten most discriminative numerical variables per time phase sorted by effect size defined as e = 2|AUROC-0.5| and calculated via univariate Mann-Whitney U tests on the training set.** The effect direction is indicated by (+)/ (−). *P*-values are FDR corrected with alpha = 0.05. Significant variables are included solely (all *P*-values < .001). Missing rates are reported as fraction of patients having values for a given variable from all patients. For time-resolved measurements, performance of aggregate scores is reported, where the 10[th], 50[th], and 90[th] percentiles are denoted as p10, p50, and p90, the median absolute deviation is denoted as map, and the sum across time is denoted as sum. Time invariant (TI), preoperative (T1), intraoperative (T2) and postoperative (T3) variables are included.

| Variable | 2\|AUROC-0.5\| | AUC | Missing rate |
|---|---|---|---|
| **TI** | | | |
| Age | (+) 0.232 | 0.616 | 0.000 |
| Number of previous diagnoses | (+) 0.134 | 0.566 | 0.000 |
| History of psychiatric disorder | (+) 0.121 | 0.560 | 0.000 |
| History of unspecific delirium | (+) 0.073 | 0.536 | 0.000 |
| History of hypertension | (+) 0.068 | 0.533 | 0.000 |
| Number of previous admissions | (+) 0.068 | 0.533 | 0.000 |
| Body length | (−) 0.051 | 0.474 | 0.547 |
| History of dementia | (+) 0.043 | 0.521 | 0.000 |
| History of respiratory failure | (+) 0.043 | 0.521 | 0.000 |
| History of diabetes mellitus | (+) 0.040 | 0.519 | 0.000 |
| **T1** | | | |
| ASA status p90 | (+) 0.179 | 0.589 | 0.476 |
| Metabolic equivalents p50 | (−) 0.178 | 0.410 | 0.866 |
| SpO2 p10 | (−) 0.155 | 0.422 | 0.360 |
| Hematocrit in blood p90 | (−) 0.140 | 0.430 | 0.891 |
| Calcium in blood p10 | (−) 0.139 | 0.430 | 0.890 |
| Hospitalization duration | (+) 0.129 | 0.564 | 0.000 |
| Erythrocytes in blood p10 | (-) 0.116 | 0.442 | 0.436 |
| Hemoglobin in blood p0.1 | (-) 0.108 | 0.446 | 0.431 |
| Ppeak p90 | (+) 0.100 | 0.550 | 0.844 |
| BP systolic | (+) 0.082 | 0.541 | 0.497 |
| **T2** | | | |
| Anesthesia duration | (+) 0.218 | 0.609 | 0.000 |
| Amount remifentanil p50 | (+) 0.200 | 0.600 | 0.695 |
| Surgery duration | (+) 0.183 | 0.591 | 0.000 |
| Amount remifentanil sum | (+) 0.178 | 0.588 | 0.695 |
| SEF right p50 | (−) 0.175 | 0.412 | 0.807 |
| SEF left p10 | (−) 0.173 | 0.414 | 0.807 |
| PCV ventilation therapy duration | (+) 0.166 | 0.582 | 0.000 |
| Endotracheal tube access duration | (+) 0.152 | 0.575 | 0.000 |
| Hospitalization duration | (+) 0.146 | 0.573 | 0.000 |
| BP systolic p90 | (+) 0.143 | 0.571 | 0.217 |
| **T3** | | | |
| Aldrete score p90 | (−) 0.347 | 0.327 | 0.079 |
| Recovery room duration | (−) 0.232 | 0.384 | 0.000 |
| Anesthesia duration | (+) 0.219 | 0.609 | 0.000 |
| Surgery duration | (+) 0.183 | 0.591 | 0.000 |
| Respiratory rate p10 | (+) 0.179 | 0.589 | 0.745 |
| PCV ventilation therapy duration | (+) 0.161 | 0.580 | 0.000 |
| Endotracheal tube access duration | (+) 0.151 | 0.575 | 0.000 |
| Heart rate p10 | (+) 0.144 | 0.572 | 0.233 |
| Respiratory rate p50 | (+) 0.143 | 0.571 | 0.745 |
| Pulse map | (−) 0.139 | 0.431 | 0.518 |

**Table 5. Top 5 most discriminative categorical variables per time phase, sorted by effect size.** The effect size is defined as |log(OR)| and calculated on the training set using univariate linear logistic regression. The effect direction is indicated by (+)/ (−). Time invariant (TI), preoperative (T1), intraoperative (T2) and postoperative (T3) variables are included. The OR 95% confidence interval (CI) serves as an uncertainty estimate.

| Variable | |log(OR)| | OR | 95% CI |
|---|---|---|---|
| **TI** | | | |
| OPS history nervous system | (+) 0.86 | 2.35 | [1.95, 2.82] |
| OPS nervous system | (+) 0.82 | 2.26 | [1.93, 2.63] |
| OPS hormone system | (+) 0.75 | 2.11 | [1.64, 2.70] |
| OPS history hormone system | (+) 0.67 | 1.96 | [1.46, 2.61] |
| OPS visual organs | (−) 0.62 | 0.54 | [0.35, 0.81] |
| **T1** | | | |
| Type spinal anesthesia | (−) 2.63 | 0.07 | [0.02, 0.22] |
| Dementia | (+) 2.48 | 11.92 | [8.21, 17.3] |
| Dissociative disorder | (+) 2.01 | 7.45 | [1.24, 44.60] |
| Cognitive impairment | (+) 1.68 | 5.39 | [2.77, 10.45] |
| Parkinson disease | (+) 1.45 | 4.28 | [3.22, 5.69] |
| **T2** | | | |
| Dementia | (+) 3.33 | 27.96 | [5.42, 144.15] |
| Urine drain access complication | (+) 2.82 | 16.77 | [2.80, 100.38] |
| Amputation | (+) 2.70 | 14.91 | [3.33, 66.63] |
| Drug related disorder | (+) 2.01 | 7.45 | [1.24, 44.60] |
| Peripheral vascular disease | (+) 1.96 | 7.12 | [2.75, 18.37] |
| **T3** | | | |
| Dementia | (+) 3.11 | 22.36 | [4.09, 122.12] |
| General op complication | (+) 2.41 | 11.18 | [2.25, 55.40] |
| Drug class benzodiazepine | (+) 1.53 | 4.61 | [2.47, 8.59] |
| Peripheral arterial disease | (+) 1.01 | 2.74 | [1.45, 5.15] |
| Parkinson disease | (+) 0.98 | 2.68 | [1.42, 5.04] |

e = 0.183) and the postoperative (T3) (anesthesia duration e = 0.219, surgery duration e = 0.183) phase. In some cases, variables with relatively high effect size had high missing rate —like the 50th percentile of the right intraoperative spectral edge frequency (SEF) (e = 0.175, 0.807 missing rate).

As seen in Table 5, dementia is the categorical variable with the highest positive association with POD encoded as EHR for all three timelines T1-T3 (o = 2.48, o = 2.48, o = 3.33). Uncertainty according to the 95% confidence interval (CI) calculated with the odds ratio [42] was very high for this variable. OPS surgical procedure history regarding the nervous system (o = 0.86), the absent application of spinal anesthesia (o = 2.63), urine drain access complication (o = 2.82) as well as general op complication (o = 2.41) are strong discriminative factors within TI, TL1-TL3 respectively.

## Model evaluation

Fig 3 summarizes the POD prediction performance of models on the test set according to the AUROC and AUPRC metrics. The upper two graphs display MLP and tree-based model performances across all time phase combinations (T1-T123). Performance was highest for models M3, M23 and M123 taking postoperative data (T3) into account. Models M12–M123 incorporating data from multiple time phases seemed to perform better than models M1–M2 focusing

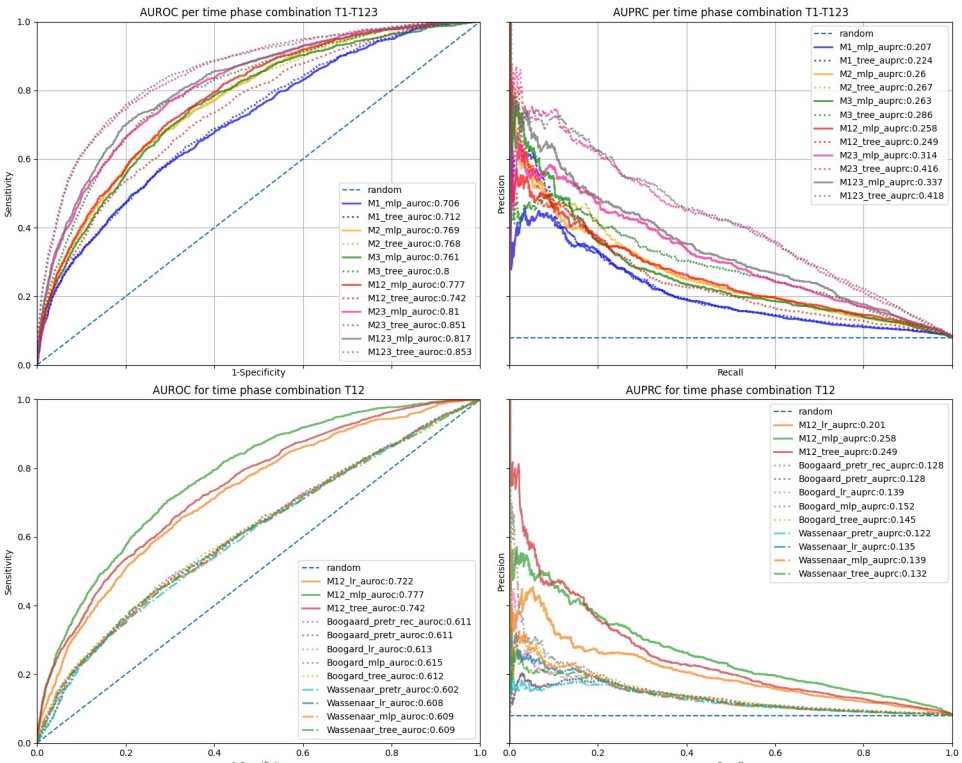

**Fig 3. POD classification performance of different models according to the area under the receiver operating characteristics curve (AUROC), and the area under the precision-recall curve (AUPRC) calculated on the test set.** Metrics are evaluated either for the MLP or tree model per variant (M1-M123, corresponding to time phases T1-T123, upper graphs) or per machine learning model class applied only to the intraoperative phase (T12, lower graphs). Every model variant includes time-invariant data (TI). Referenced baseline models are indicated by 1[st] author's name— Wassenaar or Boogaard—as prefix, recalibrated models are indicated as rec. Baseline models were either pre-trained (pretr), retrained using logistic regression—(lr) or retrained using a multi-layer perceptron (mlp).

on one single phase. Except for the combined pre- and intraoperative phase (T12), tree-based models outperformed MLPs. Tree-based model M123 ingesting all perioperative data (T123) showed highest AUROC as well as AUPRC metrics (see Fig 3).

Prediction performance of MLP, LR, tree, and baseline models—adopted from Wassenaar et al. and Boogaard et al.–applied to pre- and intraoperative data (T12), are shown in the two lower graphs of Fig 3. The proposed MLP model was superior to the linear LR model as well as the retrained or applied reference models. The best reference model for T12 was the retrained MLP model based on Boogard et al. (see Fig 3). Figs C and D in S2 Appendix display AUROC and AUPRC graphs for all model variants, baselines, and time phase combinations.

Table 6 summarizes further evaluation metrics for models per time phase combination (T1-T123) evaluated on the bootstrapped test set. The best baseline models (for Wassenaar or Boogaard) are presented as well. Non-linear tree-based models outperformed linear LR across all time phase combinations (T1-T123). Tree-based models showed higher evaluation metrics than MLP models except for one time phase (T12) (see Table 6). Non-linear MLPs outperformed linear LR with respect to the AUPRC metric for all time phases (T1-T123). Both non-linear models variants—MLPs and trees—clearly outperformed baseline models.

The best performing non-linear boosted trees variant M123 was trained on all perioperative data (T123). This model showed a mean AUROC of 0.854 (95% CI [0.853, 0.855]) and a mean

**Table 6. Performance metrics (mean, [95% confidence interval]) on bootstrapped test sets for trained logistic regression (lr), multi-layer-perceptron (mlp), tree-based models (tree), or pre-trained (pretr) models.** The best baseline models according to the AUROC and AUPRC metrics (Wassenaar and Boogaard) are also included. Sensitivity and specificity are calculated for the threshold that maximizes their sum. Precision is calculated for the highest threshold for which recall > 0.70. Model variants (M1-M123) consume data from time phases and their combinations T1-T123. Data from TI is included for every model.

| Model | AUROC | AUPRC | Sensitivity | Specificity | Precision | F1-Score |
|---|---|---|---|---|---|---|
| **T1** | | | | | | |
| M1_lr | 0.698, [0.697, 0.699] | 0.181, [0.180, 0.182] | 0.615, [0.609, 0.621] | 0.677, [0.670, 0.683] | 0.125, [0.124, 0.125] | 0.231, [0.230, 0.233] |
| M1_mlp | 0.708, [0.706, 0.709] | 0.206, [0.205, 0.207] | 0.582, [0.574, 0.589] | 0.713, [0.706, 0.720] | 0.126, [0.125, 0.127] | 0.239, [0.238, 0.241] |
| M1_tree | 0.715, [0.714, 0.716] | 0.224, [0.223, 0.226] | 0.618 [0.612, 0.624] | 0.691, [0.684, 0.697] | 0.128, [0.127, 0.128] | 0.240, [0.238, 0.241] |
| Boogard_mlp | 0.610, [0.610, 0.611] | 0.146, [0.145, 0.146] | 0.438, [0.437, 0.439] | 0.740, [0.740, 0.741] | 0.098, [0.098, 0.098] | 0.203, [0.201, 0.203] |
| Wassenaar_mlp | 0.610, [0.609, 0.611] | 0.139, [0.138, 0.141] | 0.452, [0.451, 0.453] | 0.722, [0.721, 0.723] | 0.097, [0.096, 0.098] | 0.198, [0.198, 0.199] |
| **T2** | | | | | | |
| M2_lr | 0.711, [0.710, 0.712] | 0.183, [0.182, 0.184] | 0.662, [0.656, 0.668] | 0.658, [0.653, 0.664] | 0.135, [0.134, 0.136] | 0.237, [0.236, 0.239] |
| M2_mlp | 0.766, [0.766, 0.767] | 0.253, [0.252, 0.255] | 0.742, [0.737, 0.747] | 0.649, [0.644, 0.655] | 0.161, [0.160, 0.161] | 0.258, [0.256, 0.259] |
| M2_tree | 0.771, [0.770, 0.772] | 0.273, [0.272, 0.275] | 0.732, [0.728, 0.736] | 0.676, [0.672, 0.681] | 0.167, [0.167, 0.168] | 0.269, [0.268, 0.270] |
| Boogard_mlp | 0.609, [0.608, 0.611] | 0.146, [0.145, 0.147] | 0.450, [0.446, 0.454] | 0.728, [0.724, 0.732] | 0.098, [0.098, 0.099] | 0.202, [0.201, 0.203] |
| Wassenaar_mlp | 0.609, [0.608, 0.610] | 0.139, [0.138, 0.140] | 0.459, [0.452, 0.467] | 0.716, [0.709, 0.723] | 0.097, [0.097, 0.098] | 0.198, [0.197, 0.199] |
| **T3** | | | | | | |
| M3_lr | 0.768, [0.767, 0.769] | 0.238, [0.237, 0.239] | 0.692, [0.688, 0.697] | 0.715, [0.710, 0.719] | 0.169, [0.167, 0.170] | 0.279, [0.278, 0.280] |
| M3_mlp | 0.763, [0.762, 0.764] | 0.254, [0.252, 0.255] | 0.784, [0.782, 0.786] | 0.629, [0.627, 0.631] | 0.163, [0.163, 0.164] | 0.259, [0.259, 0.260] |
| M3_tree | 0.799, [0.799, 0.800] | 0.285, [0.284, 0.287] | 0.740, [0.737, 0.743] | 0.741, [0.738, 0.744] | 0.211, [0.210, 0.212] | 0.315, [0.313, 0.316] |
| Boogard_mlp | 0.606, [0.605, 0.607] | 0.137, [0.136, 0.138] | 0.449, [0.445, 0.454] | 0.723, [0.719, 0.728] | 0.097, [0.096, 0.097] | 0.197, [0.196, 0.198] |
| Wassenaar_mlp | 0.609, [0.608, 0.61] | 0.135, [0.134, 0.136] | 0.442, [0.436, 0.448] | 0.736, [0.731, 0.741] | 0.098, [0.097, 0.098] | 0.201, [0.20, 0.202] |
| **T12** | | | | | | |
| M12_lr | 0.722, [0.721, 0.723] | 0.201, [0.200, 0.202] | 0.639, [0.633, 0.644] | 0.695, [0.691, 0.700] | 0.137, [0.136, 0.138] | 0.249, [0.248, 0.251] |
| M12_mlp | 0.777, [0.776, 0.778] | 0.269, [0.268, 0.271] | 0.760, [0.752, 0.767] | 0.652, [0.644, 0.659] | 0.167, [0.167, 0.168] | 0.265, [0.263, 0.267] |
| M12_tree | 0.740, [0.739, 0.741] | 0.246, [0.244, 0.247] | 0.651, [0.644, 0.659] | 0.696, [0.689, 0.704] | 0.142, [0.141, 0.143] | 0.255, [0.253, 0.256] |
| Boogard_tree | 0.613, [0.612, 0.613] | 0.139, [0.138, 0.139] | 0.486, [0.485, 0.487] | 0.698, [0.697, 0.699] | 0.101, [0.100, 0.102] | 0.202, [0.201, 0.203] |
| Wassenaar_mlp | 0.609, [0.608, 0.609] | 0.151, [0.15, 0.152] | 0.663, [0.659, 0.667] | 0.598, [0.594, 0.601] | 0.121, [0.121, 0.122] | 0.214, [0.214, 0.215] |
| **T23** | | | | | | |
| M23_lr | 0.751, [0.750, 0.752] | 0.227, [0.226, 0.229] | 0.708, [0.703, 0.713] | 0.676, [0.672, 0.681] | 0.159, [0.159, 0.160] | 0.262, [0.260, 0.263] |
| M23_mlp | 0.816, [0.815, 0.817] | 0.341, [0.339, 0.342] | 0.767, [0.764, 0.770] | 0.728, [0.726, 0.730] | 0.216, [0.214, 0.217] | 0.314, [0.313, 0.315] |

(*Continued*)

**Table 6.** (Continued)

| Model | AUROC | AUPRC | Sensitivity | Specificity | Precision | F1-Score |
|---|---|---|---|---|---|---|
| M23_tree | 0.851, [0.850, 0.852] | 0.410, [0.408, 0.412] | 0.778, [0.773, 0.782] | 0.779, [0.774, 0.783] | 0.277, [0.275, 0.279] | 0.362, [0.359, 0.365] |
| Boogard_mlp | 0.616, [0.615, 0.617] | 0.145, [0.144, 0.156] | 0.480, [0.475, 0.484] | 0.710, [0.706, 0.714] | 0.100, [0.100, 0.101] | 0.206, [0.205, 0.207] |
| Wassenaar_mlp | 0.608, [0.607, 0.609] | 0.137, [0.136, 0.138] | 0.445, [0.439, 0.451] | 0.730, [0.724, 0.735] | 0.097, [0.097, 0.098] | 0.199, [0.198, 0.199] |
| **T123** | | | | | | |
| M123_lr | 0.778, [0.776, 0.779] | 0.260, [0.258, 0.262] | 0.689, [0.682, 0.697] | 0.733, [0.725, 0.74] | 0.176, [0.174, 0.178] | 0.291, [0.288, 0.293] |
| M123_mlp | 0.820, [0.819, 0.821] | 0.333, [0.331, 0.336] | 0.749, [0.744, 0.754] | 0.754, [0.749, 0.76] | 0.227, [0.225, 0.229] | 0.328, [0.326, 0.331] |
| M123_tree | 0.854, [0.853, 0.855] | 0.418, [0.415, 0.421] | 0.790, [0.784, 0.796] | 0.772, [0.766, 0.778] | 0.281, [0.278, 0.283] | 0.360, [0.356, 0.363] |
| Boogard_tree | 0.616, [0.615, 0.617] | 0.150, [0.149, 0.151] | 0.450, [0.450, 0.450] | 0.724, [0.724, 0.724] | 0.098, [0.098, 0.098] | 0.198, [0.198, 0.198] |
| Wassenaar_mlp | 0.608, [0.607, 0.609] | 0.134, [0.133, 0.135] | 0.438, [0.432, 0.444] | 0.737, [0.731, 0.743] | 0.098, [0.098, 0.098] | 0.200, [0.199, 0.201] |

AUPRC of 0.418(95% CI [0.415, 0.421]]). Model variant M12, which incorporated data from the preoperative- (T1) and intraoperative (T2) phase omitting postoperative data (T3), yielded a mean AUROC of 0.777 (95% CI [0.776, 0.778]), a mean AUPRC of 0.269 (95% CI [0.268, 0.271]). Table G in S2 Appendix shows training results for tree-based models. Gradient boosted trees were evaluated as superior to random forest during all perioperative phases, except for the single postoperative phase (T3). Table H in S2 Appendix shows metrics achieved on the training dataset without a tendency of under- or overfitting. In Table J, Figs E, and F in S2 Appendix, we present calibration measures indicating a challenging model fit due to imbalanced predictions.

## Discussion

### Principal results

Our results show that non-linear models can better predict POD onset in the recovery room than linear LR models especially when ingesting features from multiple perioperative phases. Tree-based models, primarily implemented as boosted trees, outperformed MLP models in time phases T1-T123 except for T2. This observation could be explained by the selected feature set that was different for MLPs determined via cross-validation on the training data (see Tables E and F in S1 Appendix). Retrained and applied baseline models by Wassenaar and Boogaard —originally developed as delirium prediction models for the intensive care admission— yielded moderate performance in the recovery room setting.

Most of the univariate significant variables—like increased age, increased ASA score, or the presence of comorbidities—are already known from clinical studies [64–67]. It could be shown that additional parameters—like intraoperative EEG edge frequencies and procedure durations—were discriminative as well. Patients who underwent deeper sedations were more vulnerable to POD. Time-related information like prolonged anesthesia durations were also found as important predictors.

## Clinical relevance

In clinical practice, it is desirable to know the risk for POD at the end of the intraoperative phase. This knowledge could be used to initiate preventive measures such as transportation to a noise-reduced ward after the surgery [10,29,30]. Models ingesting T12 make predictions before the admission to a recovery room. Thus, the physician can decide to transfer the patient to a specialized ward [31].

Assuming 100 surgeries per day through all three hospital sites including 10 real cases of POD. The application of the MLP M12 with a fixed sensitivity at 0.80 and a corresponding precision of 0.20 would lead to 8 correct transfers—of patients really suffering from POD—and 32 incorrect transfers—of patients not suffering from POD—to a specialized ward after surgical procedures. With a usual ICU size of 15–20 patients, the results highlight that a low precision is a main impediment for implementing trained models in a real clinical setting. We provide a configurable prediction model for balancing overtreatment and patient safety.

We retrieved MLP input weights (w) for 441 features including missing indicators recorded for the time phase T12. We identified age (w = 0.907), history of delirium (w = 0.784), and history of hypertension (w = 0.631) as important model inputs. These variables describe predisposing factors that might contribute to illness severity increasing the chance of POD. We observed elevated model weights regarding the binary missing indicator for intraoperative pulse (w = 0.526) and sedation depth (w = 0.379). Observations suggest that POD predictions can benefit from encoded availability information. The median-aggregated amount of remifentanil given during the intraoperative phase (T2) yielded a weight of 0.337. The predictor showed a positive POD correlation in univariate analyses in line with guidelines that suggest opioid sparse sedations [10,11]. In Table F in S1 Appendix, we present further model weights. Results were complementary with univariate findings.

## Comparison with related work

Original work by Wassenaar et al. and Boogard at al. focused on delirium prediction based on data available early after admission [23,68]. We could achieve a maximum AUROC of 0.61 using preoperative data by retraining their models, which seems to be complementary to external validation studies [24]. In our setup, the highest observed AUROC achieved for this time phase was 0.715. Xue et al. combined pre- and intraoperative data for training a MLP, reporting an AUROC of 0.715 and an AUPRC of 0.731 on data with 52.6% prevalence [25]. MLP model M12, which was also trained on pre- and intraoperative data, achieved a similar mean AUROC value (0.777). Due to the reduced prevalence, the mean AUPRC was noticeably lower (0.269). Low POD prevalence was explicitly addressed by Davoudi et al. using oversampling [6], we wanted to train an applicable model without changing the prevalence. Davoudi et al. and Bishara et al. achieved promising AUROC values over 0.80 with non-linear models on preoperative data, but did not report any AUPRC metrics [6,27]. This would have been beneficial for a comprehensive comparison. Racine et al. compared a linear LR with a MLP approach. Their MLP model achieved an AUROC of 0.71 and a linear LR model achieved an AUROC of 0.69 [26]. Most related work investigating the application of machine learning for POD prediction with linear LR during a clinical trial incorporating few samples and specialized attributes [16–20]. Scores explicitly designed for assessing cognitive impairments—like the Mini-Mental State Examination (MMSE)–are highly correlated with delirium and were included in these studies as predictor variables [16,26]. Positive ICD or CAM values during hospitalization were used by referenced work for the target definition [23–26]. To our knowledge, no previous study focused on a Nu-DESC POD assessment in a recovery room setting.

## Strengths

We systematically investigated the contribution of aggregated clinical variables to the prediction of POD under varying perioperative time phases. We included pre-, intra-, and postoperative data. The combination of features from multiple time phases improved predictions made by highly non-linear machine learning models. We extracted model weights for providing insights into our best performing applicable MLP model M12. Results were accompanied by robust univariate test statistics revealing effects of single predictors towards our POD target variable. Our models are openly accessible and can potentially be evaluated in other medical centers.

## Limitations

Due to the vast amount of records, there was no chance of ensuring clinical correctness for all extracted EHRs. We rather ensured that extracted variables reflected our front-end systems. Our univariate analysis and prediction models based on clinical data that could potentially comprise extreme values (e.g. body temperature of 44.3 ˚C) due to flawed documentations. We believe that our models benefited from these noisy data by using robust feature aggregations—like percentiles—increasing model's applicability. However, univariate statistical results must be clinically interpreted carefully. No cohort matching—e.g., for reducing cofounding bias in a case-control study—was done. The feature selection process moreover ignored dependencies between covariates, which may have been beneficial for the predictive performance [69]. Some features also showed high predictive power but low availability. We encoded feature availability as missing indicators that could leverage POD predictions.

We did not use feature interpretation methods—such as LIME or SHAP—as such methods are themselves poorly understood and may lead to wrong conclusions about model and data [69,70]. Since we applied $L_1$-norm regularization on the first MLP layer, we could retrieve model input weights. We believe that these findings should contribute to further clinical validation steps instead of directly impacting clinical practice due to the high complexity of ML models. We conducted a single-center study, results could have benefited from external validation. Many openly accessible clinical research databases comprise ICU data instead of perioperative clinical records hampering the validation of our results [71,72]. We also limited our target variable to Nu-DESC observations that might not be available in other clinical centers. Our cohort definition excluded patients undergoing cardiac—or craniotomy surgical procedures.

Models incorporating T3 provide a POD assessment without relying on the actual observed Nu-DESC. Results were displayed to assess the relevance of covariates measured into the recovery room. We focused our clinical interpretation solely on MLP model M12 ingesting data up to T2. Cases with later POD onsets in ICUs or cases bypassing the recovery room were ignored but can be investigated in further studies. A prospective study that validates the predictions of our models also focusing on a clinical assessment regarding the Nu-DESC would be beneficial towards a clinical application. We identified a low prevalence of 8.2% as a main impediment for a well-calibrated model fitting.

## Conclusion

This study demonstrates that machine learning can be used to predict POD assessed by the Nu-DESC in the recovery room, where the incorporation of different intraoperative phases as feature sets proved useful. Overall, non-linear models were superior to linear LR techniques as well as known published models. We presented the clinical impact of one model configuration outbalancing patient safety and overtreatment. Results could guide decision makers that are

eager to explore machine learning for mitigating POD risks. However, strategies for highly imbalanced data must be developed to implement solutions in clinical practice.

## Supporting information

**S1 Appendix. Additional tables comprising data for feature encodings, model parameters, and selected feature sets.**
(XLSX)

**S2 Appendix. Additional tables and figures for cohort characteristics, model training procedure, and guideline adherence.**
(DOCX)

## Author Contributions

**Conceptualization:** Niklas Giesa, Stefan Haufe, Björn Weiß, Claudia D. Spies, Felix Balzer, Sebastian D. Boie.

**Data curation:** Mario Menk.

**Formal analysis:** Niklas Giesa, Stefan Haufe.

**Methodology:** Niklas Giesa, Sophie K. Piper.

**Validation:** Niklas Giesa.

**Visualization:** Niklas Giesa.

**Writing – original draft:** Niklas Giesa.

**Writing – review & editing:** Niklas Giesa.

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
