## [Decision Letter · Decision Letter 0]

5 Apr 2024

PDIG-D-23-00435

Predicting postoperative delirium assessed by the Nursing Screening Delirium Scale in the recovery room for non-cardiac surgeries without craniotomy: A retrospective study using a machine learning approach

PLOS Digital Health

Dear Dr. Giesa,

Thank you for submitting your manuscript to PLOS Digital Health. After careful consideration, we feel that it has merit but does not fully meet PLOS Digital Health's publication criteria as it currently stands. Therefore, we invite you to submit a revised version of the manuscript that addresses the points raised during the review process.

Please submit your revised manuscript within 60 days Jun 04 2024 11:59PM. If you will need more time than this to complete your revisions, please reply to this message or contact the journal office at digitalhealth@plos.org. Please include the following items when submitting your revised manuscript:

We look forward to receiving your revised manuscript.

Kind regards,

Nicole Yee-Key Li-Jessen

Academic Editor

PLOS Digital Health

Journal Requirements:

Additional Editor Comments (if provided):

Reviewers' comments:

Reviewer's Responses to Questions

**Comments to the Author**

1. Does this manuscript meet PLOS Digital Health’s publication criteria? Is the manuscript technically sound, and do the data support the conclusions? The manuscript must describe methodologically and ethically rigorous research with conclusions that are appropriately drawn based on the data presented.

Reviewer #1: Yes

Reviewer #2: Yes

Reviewer #3: No

2. Has the statistical analysis been performed appropriately and rigorously?

Reviewer #1: I don't know

Reviewer #2: Yes

Reviewer #3: No

3. Have the authors made all data underlying the findings in their manuscript fully available (please refer to the Data Availability Statement at the start of the manuscript PDF file)?

Reviewer #1: No

Reviewer #2: Yes

Reviewer #3: No

4. Is the manuscript presented in an intelligible fashion and written in standard English?

Reviewer #1: Yes

Reviewer #2: Yes

Reviewer #3: No

5. Review Comments to the Author

Reviewer #1: In the manuscript "Predicting postoperative delirium in the recovery room", the authors constructed a predictive model for postoperative delirium using clinical information from a large cohort of patients. I have several issues:

1. Did any patients in the study undergo multiple surgeries during the observation period? For such patients, were all surgical records included in the study, or only records from the initial surgery?

2. The appendix mentions,“A feature selection algorithm was devised to provide a tradeoff between data availability and predictive power towards the target.” Is this algorithm designed for variable selection? Or are you including all variables as predictor variables in the model? If feature selection was performed, please provide a detailed explanation of the selection method and the predictor factors ultimately included in the models.

3. In the methods section，it mentions“To avoid dependencies between these sets we used patient identifiers to perform the splitting. Stratification with the target variable was done so that the incidence of POD was preserved in both sets.” Was the training set and validation set not divided randomly? Were there differences in the variables between the training set and validation set？

4. In the methods section，it mentions “Extracted training set mean values were also used to impute missing values in train, validation, and test sets.” What is the reason for having three datasets?

5.In the table 3, is the definition of the missing rate accurate? And in the addendix, “all numerical features that were not available for at least 10% of the patients were dropped”, however, there are many features in the table3 with a missing rate higher than 10%, please explain this. Additionally, would the presence of numerous features with high missing rates affect the model's outcomes? Would it compromise the credibility of the model?

6.In table 3, “Highly correlated clinical variables were age (e=0.232, with e=2|AUROC-0.5|) for TI, the ASA status (e=0.179) for T1, the intraoperative (T2) amount of remifentanil (e=0.200), and the Aldrete score (e=0.347) measured in the recovery room (T3).”Why isn't it Anesthesia duration (e=0.218)? This seems inconsistent with the results in Table 3.

7. please provide a detailed explanation of the predictor variables in the best model

8. The authors need to adhere to STROBE guidelines. There are numerous other requirements for adequate reporting of a study.

9. Please specify the strengths of this study in the discussion section.

Reviewer #2: This study significantly contributes to the field by applying ML to predict POD in a recovery room setting. The methodology is rigorous, and the findings have clear implications for improving patient outcomes post-surgery. Including pre-, intra-, and postoperative data in the ML models is a notable strength. The paper dealt with rare events. The author used specific metrics for rare event prediction, and both AUROC and AUPRC metrics thoroughly evaluate model performance. However, the paper could be strengthened in the following aspects:

1. The introduction might be enhanced by more explicitly stating the study’s novel contributions beyond the application of ML, particularly how it advances understanding of POD risk factors or patient outcomes.

2. While the methods are generally strong, the paper could provide more detail on handling missing data and justifying the choice of ML models over others.

3. Would over-sample or down-sample further help predict the “rare event”? 

4. Which tree method gave the best performance? Random forest or boosting trees? 

5. The study's single-center nature may limit the generalizability of the findings. Is it possible to find an external validation cohort?

6. The results section could benefit from a deeper analysis of which specific features contributed most to the prediction models and why. 

7. Additionally, discussing the clinical relevance of prediction accuracy in practical settings would be beneficial.

8. Calibration metrics were not provided. 

9. The discussion of limitations is brief. A more detailed exploration of the study's limitations, including its retrospective design and potential for bias, would strengthen the paper. 

10. The conclusion could address future research directions more directly and how these findings could influence clinical practice or policy.

Reviewer #3: In the author summary paragraph, I suggest rewording and defining “mentally disturbed”. For example, consider “Many patients who experience POD suffer from chronic mental health disorders…”. Also consider rewording “With our study, we want to detect POD before...” to “With our study, we want detect suspected POD”. 

“Vulnerable patients could be transferred to delirium wards mitigating the risk of severe outcomes such as permanent cognitive decline.” Are there delirium wards in Germany? Consider saying “transferred to a higher level of care…” Also question the use of “permanent cognitive decline”. Consider using “long-term cognitive decline”.

While the etiology of POD is difficult to prove conclusively, Li et al. (2022) make the case that while surgical stress can lead to local inflammation and immune activation, it can also lead to a systematic cascade of inflammatory signaling molecules which induce, maintain, and aggravate neuroinflammation and subsequent POD, especially in the elderly population. This makes sense as Andronie-Cioara et al. (2023) report that aging is associated “with a chronic inflammatory state both in the periphery and in the central nervous system”. Therefore the activation of the inflammatory cascade during a surgical procedure in patients with chronic neuroinflammation is a viable explanation for POD in elderly and in patients with other underlying chronic inflammatory conditions. 

Li Z, Zhu Y, Kang Y, Qin S, Chai J. Neuroinflammation as the Underlying Mechanism of Postoperative Cognitive Dysfunction and Therapeutic Strategies. Front Cell Neurosci. 2022;16:843069. Published 2022 Mar 28. doi:10.3389/fncel.2022.843069

Andronie-Cioara FL, Ardelean AI, Nistor-Cseppento CD, et al. Molecular Mechanisms of Neuroinflammation in Aging and Alzheimer's Disease Progression. Int J Mol Sci. 2023;24(3):1869. Published 2023 Jan 18. doi:10.3390/ijms24031869

What does ASA status in the abstract refer to? Is it American Society of Anesthesiologists (ASA) Physical Status Classification System? What is “OPS” surgical procedure history (page 10)? All acronyms need to be spelled out prior to use.

In Table in the supplement the lower bounds for GCS is listed as 0. The lowest possible GCS score is 3. Are 0 values missing values? What imputation methods were used and how did the researchers determine if missingness was structurally missing, missing completely at random (MCAR), missing at random, or missing not at random?

I also question the vital sign lower bounds (all were 0 except temperature) and even the upper bounds were not conducive with life. For example, a temperature of 10 Celsius (50 Fahrenheit) or 50 Celsius (122 Fahrenheit) are not conducive with life and are likely artifacts. How did researchers manage these artifacts. The same goes for measurements of 0 for respiratory rate, heart rate, blood pressure… My assumption is that these were either missing values or artifacts. How were these managed? I have never seen a patient on a tidal volume of 10,000 ml. I have similar concerns with lower/upper limits in other variables in this table.

Description of Table 1 in the supplementary Excel Spreadsheet: Extended Table 1: Hard rules in terms of ranges from valid lower - to upper bounds for data preprocessing. Values that do not fall into the valid range were imputed with the mean calculated on the training set.

In the Extended Table 3 of this same spreadsheet seems to report high rates of missingness as well as some min and max values not conducive with life. See example below.

feature set domain unit tl missingness_rate min max

body temperature train Vital Signs °C 2 0.44 10.90 45.40

body temperature train Vital Signs °C 1 0.96 10.30 44.30

body temperature train Vital Signs °C 3 0.99 12.90 39.80

body temperature test Vital Signs °C 2 0.45 16.20 39.90

body temperature test Vital Signs °C 1 0.95 10.20 39.90

body temperature test Vital Signs °C 3 0.99 18.30 38.30

Lastly, there was limited demographic information (only included age, BMI, and gender) so it is difficult to determine generalizability. 

The researchers need to better explain how they prepared data for analysis. Did homologation occur e.g. checking the mapping of the data for correctness, consistency, and possible corruption after the ETL process? Did data corruption occur during the ETL process (resulting in deletion or modification of information, inclusion of extraneous or confounding data, and misinterpretation of the clinical meaning of data)? If those steps were not feasible then how were the meanings of variables determined/managed?

6. PLOS authors have the option to publish the peer review history of their article (what does this mean?). If published, this will include your full peer review and any attached files.

**Do you want your identity to be public for this peer review?** For information about this choice, including consent withdrawal, please see our Privacy Policy.

Reviewer #1: No

Reviewer #2: No

Reviewer #3: No

---

## [Decision Letter · Decision Letter 1]

4 Jul 2024

Predicting postoperative delirium assessed by the Nursing Screening Delirium Scale in the recovery room for non-cardiac surgeries without craniotomy: A retrospective study using a machine learning approach

PDIG-D-23-00435R1

Dear Mr Giesa,

We are pleased to inform you that your manuscript 'Predicting postoperative delirium assessed by the Nursing Screening Delirium Scale in the recovery room for non-cardiac surgeries without craniotomy: A retrospective study using a machine learning approach' has been provisionally accepted for publication in PLOS Digital Health.

Best regards,

Nicole Yee-Key Li-Jessen

Academic Editor

PLOS Digital Health

Reviewer Comments (if any, and for reference):

Reviewer's Responses to Questions

**Comments to the Author**

1. If the authors have adequately addressed your comments raised in a previous round of review and you feel that this manuscript is now acceptable for publication, you may indicate that here to bypass the “Comments to the Author” section, enter your conflict of interest statement in the “Confidential to Editor” section, and submit your "Accept" recommendation.

Reviewer #3: All comments have been addressed

2. Does this manuscript meet PLOS Digital Health’s publication criteria? Is the manuscript technically sound, and do the data support the conclusions? The manuscript must describe methodologically and ethically rigorous research with conclusions that are appropriately drawn based on the data presented.

Reviewer #3: Yes

3. Has the statistical analysis been performed appropriately and rigorously?

Reviewer #3: Yes

4. Have the authors made all data underlying the findings in their manuscript fully available (please refer to the Data Availability Statement at the start of the manuscript PDF file)?

Reviewer #3: No

5. Is the manuscript presented in an intelligible fashion and written in standard English?

Reviewer #3: Yes

6. Review Comments to the Author

Reviewer #3: This is a well-designed study and well-written paper. Thank you for your thorough response to all of my comments and questions.

7. PLOS authors have the option to publish the peer review history of their article (what does this mean?). If published, this will include your full peer review and any attached files.

**Do you want your identity to be public for this peer review?** For information about this choice, including consent withdrawal, please see our Privacy Policy.

Reviewer #3: **Yes: **Teresa Rincon
